# Carbapenemase-Producing Bacteria Isolated from ICU Patients of a Peruvian Government Hospital during the COVID-19 Pandemic: A Descriptive Analysis

**DOI:** 10.3390/medicina59101763

**Published:** 2023-10-03

**Authors:** David García-Cedrón, Magaly De La Cruz Noriega, Luis Cabanillas-Chirinos, Nélida Milly Otiniano, Walter Rojas-Villacorta, Waldo Salvatierra-Espinola, Karen Diaz Del Aguila, Manuela Luján-Velásquez

**Affiliations:** 1Facultad de Ciencias de la Salud, Universidad César Vallejo, Trujillo 13001, Peru; dcgarciac@ucvvirtual.edu.pe (D.G.-C.); wsavatierraes@ucvvirtual.edu.pe (W.S.-E.); 2Vicerrectorado de Investigación, Universidad Autónoma del Perú, Lima 15842, Peru; mdelacruzn@autonoma.edu.pe; 3Instituto de Investigación en Ciencia y Tecnología, Universidad César Vallejo, Trujillo 13001, Peru; lcabanillas@ucv.edu.pe (L.C.-C.); notiniano@ucv.edu.pe (N.M.O.); 4Programa de Investigación Formativa, Universidad César Vallejo, Trujillo 13001, Peru; kdiazd@ucv.edu.pe; 5Facultad de Ciencias Biológicas, Universidad Nacional de Trujillo, Trujillo 13001, Peru; mlujan@unitru.edu.pe

**Keywords:** pandemic, COVID-19, carbapenemases, antimicrobial resistance, antimicrobial use

## Abstract

*Background and Objectives:* In Peru, the presence of antimicrobial-resistant bacteria is a constant concern in hospitals and has likely increased in frequency during the pandemic. The objective of the study was to analyze the frequency of carbapenemase-producing bacteria resistant to two carbapenems (Imipenem and Meropenem), which were isolated from Peruvian patients in the intensive care unit of the Victor Lazarte Echegaray Hospital in Trujillo (Peru) during the COVID-19 pandemic. *Materials and Methods:* The biological samples of the patients hospitalized in the ICU were processed in the Microbiology Diagnostic Laboratory of the Víctor Lazarte Echegaray Hospital between May 2021 and March 2022. Antimicrobial sensitivity was determined with the automated system AutoScan-4, and for the identification of the type of carbapenemase, the RESISIT-3 O.K.N K-SET cassettes were used. *Results:* The results show that 76 cultures (76/129) had resistance to the two carbapenems (imipenem or meropenem), where the most frequent were *Klebsiella pneuomoniae* (31.6%), *Pseudomonas aeruginosa* (26.3%), and *Acinetobacter baumannii* (14.5%). *Pseudomonas aeruginosa* cultures showed at least three carbapenemase types (KPC, NDM, and OXA-48), while *A. baumannii*, *Escherichia coli*, and *Burkholderia cepacia complex* presented at least two carbapenemases (NDM and OXA-48). The carbapenemase NDM was detected in *Enterobacter cloacae*, *Morganella morganii*, and *Proteus mirabilis*, while KPC was present in all *Klebsiella pneumoniae* and *Klebsiella oxytoca* cultures. *Conclusions*: The samples from patients hospitalized in the Victor Lazarte Echegaray Hospital ICU showed a high prevalence of imipenem- and meropenem-resistant bacteria. These findings are relevant and concerning from the perspective of antibiotic-resistant bacteria monitoring, control, and disinfection. Thus, an appropriate antibiotic policy must be implemented.

## 1. Introduction

The *Enterobacteriaceae* family contains the most varied and extensive group of Gram-negative bacilli (BGN), which are clinically significant because they cause a wide range of diseases in humans [1,2]. Enterobacteriaceae are rod-shaped bacilli ranging from 1 to 3 μm long and can grow in the presence and absence of oxygen (facultative anaerobes) [3,4]. These bacilli cause hospital-acquired infections and infections acquired in the community because they can cause infections in the urinary tract, respiratory tract, bloodstream, and exposed wounds [5,6,7]. The Pan American Health Organization (PAHO) and the World Health Organization (WHO) mention that only five of the seven families of *Enterobacteriales* have clinical importance, due to their relationship with human diseases. Some of the genera that are included in this family are *Salmonella*, *Shigella*, *Escherichia*, *Klebsiella*, *Enterobacter*, *Citrobacter*, *Yersinia*, *Serratia*, *Morganella*, *Proteus*, and *Providencia* [8].

In recent years, infections caused by antibiotic-resistant and multidrug-resistant bacteria have increased progressively, leading to the ineffectiveness of treatments [9]. The Spanish Society of Infectious Diseases and Clinical Microbiology (SEIMC) published a report indicating that if no effort is made to combat antibiotic resistance, more than 10 million people may die by 2050 [10]. This prediction considers the data reported in 2019, which showed approximately 4.95 million deaths worldwide due to bacterial resistance.

During the COVID-19 pandemic, PAHO/WHO issued an epidemiological alert, which reported that in Latin America and the Caribbean, there was an increase in new mixtures of carbapenemases in species of *Enterobacteriaceae*. It also mentions that this increase is possibly related to an increase in the uncontrolled use of broad-spectrum antibiotics in patients with COVID-19 [8]. In Peru, Yañez (2021) reported that COVID-19 caused an accelerated increase in the number of patients hospitalized in intensive care units (ICUs). At the same time, there was evidence of irrational use of antimicrobials without pharmacological support, resulting in a rise of bacteria resistant to various antimicrobials [11,12].

Bacterial infections are managed and treated with beta-lactam antibiotics. These are the most widely used types of drugs globally. Nowadays, the annual expenditure on these antibiotics amounts to approximately USD 15,000 million and represents 65% of the total antibiotic market [13]. One class of beta-lactams is carbapenems, which have a broad spectrum of antibacterial activity against Gram-positive and Gram-negative microorganisms [14]. These antibiotics are considered to be among the last-line drugs and the most reliable for treating bacterial infections. They are also safer than other similar drugs because they have few adverse side-effects [15]. Carbapenems interrupt bacterial cell wall formation as a result of covalent binding to the essential penicillin-binding proteins 1a, 1b, and 3 (PBP1a, 1b, and 3), which induces spheroplast formation and cell lysis without filament formation [16,17].

Until 2010, the drugs imipenem, meropenem, ertapenem, and doripenem were the most used carbapenems in clinical practice for the treatment of infectious diseases caused by bacteria [18]. Of these carbapenems, imipenem and meropenem possess the broadest antibacterial spectrum [19]. The first carbapenem utilized was imipenem in 1985. Meropenem is now the most widely used antibiotic since it can treat a wide range of respiratory, intra-abdominal, and nosocomial infections, such as infections caused by *Pseudomonas aeruginosa* [20,21]. Imipenem and meropenem are usually highly resistant to hydrolysis by most clinically important beta-lactamases; both carbapenems inhibit bacterial wall synthesis, similarly to other β-lactam antibiotics. The fundamental difference between these carbapenems is evident in terms of their mechanism of action [1,17,18].

Overuse of beta-lactam antibiotics can lead to resistance mechanisms in bacteria. This resistance can occur through different mechanisms, including target modification (mutation or expression of alternative PBPs), cell permeability reduction through downregulation of porins required for beta-lactam entry, over-expression of efflux systems, and the production of modifying or degradative enzymes [22]. The increase in bacterial resistance is also due to the presence of enzymes called beta-lactamases, which inhibit the mechanism of action of β-lactam antibiotics through a hydrolysis process [19,22,23].

Multidrug-resistant Gram-negative bacteria constitute a public health problem of global relevance; because of this, the PAHO/WHO published a list indicating the need to look for new antibiotics for carbapenems-resistant enterobacteria or producers of extended-spectrum beta-lactamases due to the increase in resistance of BGNs to carbapenems worldwide [1,8]. Initially, resistance to carbapenem was mainly found in *Pseudomonas aeruginosa* and *A. baumannii*, but in recent decades, *Klebsiella pneumoniae* has come to be recognized as another bacterium exhibiting resistance [24]. 

In 2019, the National Reference Laboratory of Hospital-Acquired Infections (LRNIIH) of the National Institute of Health (INS) carried out a study wherein it molecularly characterized the carbapenemases present in cultures from the twelve regions of Peru, identifying the following types of carbapenemases, along with the co-production of *bla*_VIM/IMP_: *bla*_KPC_, *bla*_NDM_, *bla*_IMP_, *bla*_VIM_, *bla*_OXA-23_, *bla*_OXA-24_, and *bla*_OXA-51_, [25]. In 2020, the Ministry of Health of Peru (MINSA) reported in the Lambayeque region the discovery of the first strain of *A. baumannii* of the NDM + OXA-58 type. In addition, in a hospital in Lima, two strains of *Klebsiella pneumoniae* were identified, with double production of carbapenemases (KPC + NDM), while in the Arequipa region, an unusual strain of *Escherichia coli* with double production of carbapenemases (NDM + OXA-48) was detected [11,26].

Several studies show that, within hospitals, the intensive care unit (ICU) is an area where the risk of antimicrobial resistance is high due to the high rate of antibiotic use [27,28]. Patients with infections caused by drug-resistant microorganisms are frequently admitted to the ICU, and the percentage of bacteria identified in ICUs with antibiotic resistance is nearly double that found in other clinical settings [29].

Nowadays, antibiotic resistance in microorganisms has resulted in various novel containment approaches for multidrug-resistance bacteria. For this reason, it is necessary to conduct studies and research that help identify phenotypically multidrug-resistant microorganisms in distinct areas of the hospital. Therefore, the present study aimed to determine the frequency and type of carbapenemases present in Gram-negative bacteria isolated from patients hospitalized in the ICU of the Victor Lazarte Echegaray Hospital, located in the city of Trujillo (Peru), during the COVID-19 pandemic.

## 2. Materials and Methods

### 2.1. Design of the Study

A descriptive, observational cross-sectional study was conducted between May 2021 and March 2022 in the ICU of Víctor Lazarte Echegaray Hospital, located in Trujillo, Peru, where a large number of patients with bacterial infections are treated.

### 2.2. Population and Sample 

The population consisted of 129 bacterial cultures isolated from samples from ICU patients of the Víctor Lazarte Echegaray Hospital who stayed between May 2021 and March 2022. The sociodemographic characteristics of patients are presented in Table 1. We worked with 76 cultures resistant to meropenem or imipenem. These 76 cultures consisted of 45 *Enterobacteria*, 20 *Pseudomonas aeruginosa*, and 11 *Acinetobacter baumannii*. The samples included bronchial secretions, aseptic urine, blood, central venous catheters, wound secretions, and other biological fluids. Samples from other parts of the hospital (e.g., pediatric and emergency ICUs, outpatient clinics) and non-viable cultures were excluded because they are difficult to identify using the MicroScan™ system. Finally, both axenic and viable cultures from ICU samples were considered.

### 2.3. Collection and Processing of Biological Samples

Biological samples of bronchial secretions, aseptic urine, blood, central venous catheters, wound secretions, and other biological fluids were collected and processed following the guidelines of the Manual of Laboratory Procedures: Local Laboratories I: Local Laboratories II of the Ministry of Health/National Institute of Health [30]. 

### 2.4. Selection, Identification, and Sensitivity of Bacteria

The colonies that showed growth in the plates with MacConkey agar were selected and identified through the automated system MicroScan™ AutoScan-4 (Siemens Healthcare Diagnostics Inc., St. Paul, MN, USA). Antimicrobial identification and susceptibility were performed on panels for Gram-negative bacteria, taking into account the Neg Entero Combo Panel Type 72 (Siemens Healthcare Diagnostics Inc., St. Paul, MN, USA). These panels contain the carbapenems imipenem, meropenem, and ertapenem, which are used in the treatment of *Enterobacteriaceae*, *Pseudomonas aeruginosa*, and *Acinetobacter* spp. present in urine and systemic samples. The bacterial identification and sensitivity procedure included performing a bacterial suspension and transferring 100 μL into the wells of the panels, which were incubated for 16 to 18 h at 35 °C following the indications of the Procedure Manual for Gram-negative microorganisms. To identify extended-spectrum β-lactamases, the automated system had an expert alert system. An alert is emitted when the growth of bacteria is possible in the presence of cefpodoxime concentrations beyond 4 μg/mL and ceftazidime concentrations beyond 1 μg/mL, which are concentrations recommended by The Clinical and Laboratory Standards Institute CLSI [25,30,31]. 

### 2.5. Identification of Carbapenemases

For the identification of carbapenemase enzymes, a rapid diagnostic immunochromatographic technique was used. This method was used to evaluate cultures with an MIC of 2 μg/mL for at least one carbapenem, imipenem or meropenem [32,33]. In the present study, the RESISIT-3 O.K.N K-SET rapid diagnostic test, manufactured by the company Coris BioConcept [34], was used for the detection of the carbapenemases OXA-48, KPC, and NMD. In previous studies, this test has shown 96% sensitivity and 100% specificity for the detection of NDM, OXA-48, and KPC [35].

### 2.6. Data Collection and Analysis

LabPro software version 4.43 was used to analyze data and determine biotypes. Results with high probabilities (≥85%) were considered reliable, and those with low probabilities (<85%) were deemed unconfirmed [31]. In this study, no additional tests were performed to confirm low-probability identifications, and the antibiotic resistance profile was interpreted according to cut-off points recommended by the Clinical and Laboratory Standards Institute, 33rd edition [36,37]. For a better understanding of the results, graphs, and tables were generated in the Microsoft Excel program. 

## 3. Results

Table 2 shows the frequency distribution of isolated cultures and resistance to carbapenems according to the sample type. Most cultures were isolated from samples of bronchial secretions (n = 60; 46.5%), aseptic urine (n = 41; 31.8%), and blood cultures (n = 13; 10.1%). It was found 56.6% of the bacterial cultures resistant to one of the two carbapenems corresponded to bronchial secretion samples, followed by 25% corresponding to aseptic urine samples. On the other hand, a minority of carbapenem-resistant cultures were isolated from other samples that had a lower frequency.

Figure 1 shows the distribution of the bacteria identified by the automated system AutoScan-4 from the biological samples of ICU patients processed in the Microbiology Diagnostic Laboratory of the Víctor Lazarte Echegaray Hospital from May 2021 to March 2022. The highest percentages of Gram-negative bacteria isolated were found for *K. pneumoniae* (n = 24, 31.6%), *P. aeruginosa* (n = 20, 26.3%), and *A. baumannii* (n = 11, 14.5%). On the other hand, there was a low frequency of Gram-negative bacteria that could infect ICU patients, such as *E. coli* (9.2%); *K. oxytoca* and *Burkholderia cepacia complex* (n = 3, 3.9%); *E. aerogenes* and *S. marcescens* (n = 2, 2.6%); and *E. cloacae*, *C. freundii*, *M. morganii*, and *P. mirabilis* (n = 1, 1.3%)

Table 3 shows the proportion of Gram-negative bacteria identified by the automated AutoScan-4 system in aseptic urine and bronchial secretion samples (most frequent samples) from patients in the ICU. In addition, the most commonly isolated enterobacterium from bronchial secretion samples was *K. pneuomoniae* (n = 18; 42.9%). On the other hand, *P. aeruginosa* was the bacterium most commonly isolated from aseptic urine samples (n = 5; 25%); likewise, it was the second most frequent in samples of bronchial secretion (n = 12; 28.6%). Among the bacteria isolated in at least one sample (bronchial secretion) were *E. aerogenes*, *E. cloacae*, *S. marcescens*, and *P. mirabilis*, with *M. morganii* only appearing in urine aseptic samples. 

Table 4 shows the bacterial cultures and the number of sensitive and resistant cultures. The *P. aeruginosa* culture presented a great number of cultures resistant to imipenem (n = 16) and meropenem (n = 20) antibiotics. Likewise, *K. pneumoniae* cultures presented resistance to imipenem (n = 16) and meropenem (n = 17); moreover, it also had cultures with sensitivity to both antibiotics. The third group with cultures resistant to imipenem (n = 11) and meropenem (n = 11) was *A. baumannii*. It should be noted that this bacterial group did not have cultures sensitive to both antibiotics like other bacterial groups (*K. oxytoca*, *E. aerogenes*, *S. marcescens*, *E. cloacae*, *C. freundii*, and *M. morganii*), which is extremely important.

Table 5 shows a different group of bacteria and their carbapenemases (KPC, NDM, and OXA-48). These bacteria can produce nosocomial infections. All cultures of *K. pneuomoniae* had the carbapenemase KPC (n = 24). The cultures of *P. aeruginosa* presented NDM (n = 11), OXA-48 (n = 4), and KPC (n = 1) carbapenemases. Concerning *A. baumannii*, its cultures presented NDM (n = 6) and OXA-48 (n = 1) carbapenemases. The other bacterial groups demonstrated the presence of at least one type of carbapenemase (*K. pneumoniae*, *E. coli*, *E. aerogenes*, *S. marcescens*, *E. cloacae*, *M. morganii*, and *P. mirabilis*). 

## 4. Discussion

Initially, the appearance of carbapenemases in patients with bacterial infections was rare; currently, an increased number of these hydrolyzing enzymes are being generated as part of the defense mechanism of a certain group of microorganisms, especially those of the groups *Enterobacteriaceae*, *P. aeruginosa*, and *A. baumannii*. This is generating concern among clinicians and scientific research groups due to the therapeutic challenge they represent, since it has been observed that resistance to carbapenems implies resistance to other β-lactams [5,19,38].

The results obtained show a greater predominance of bacterial isolates in samples of bronchial secretion and urine, after analyzing 43 samples of bronchial secretions (56.6%) and 19 samples of aseptic urine (25.0%) during the period May 2021 to March 2022 (Table 2). This predominance in terms of a greater number of bronchial secretion samples comes from the high number of patients with acute respiratory infections during the COVID-19 health emergency, where priority was given to patients suspected of having been infected with the SARS-CoV-2 virus. These results agree with what was reported by Tranche Iparraguirre et al. (2021), who tried to explain the impact that the COVID-19 disease had on the health of people seeking care in hospitals during the first, second, and third waves of the COVID-19 pandemic. During the health emergency, priority was given to patients with acute respiratory problems and, secondly, to other patients, including those with a serious or chronic illness. Furthermore, the care provided during the pandemic by the Víctor Lazarte Echegaray Hospital was very restricted, due to the government’s provisions regarding hospital care, since priority was given to patients who presented with a diagnosis of pneumonia associated with COVID-19. During the development of the disease and treatment, a culture of bronchial secretion was performed to diagnose or rule out a bacterial infection associated with COVID-19 [39], which is why this type of sample was one of the most requested by doctors.

When analyzing isolated bacterial cultures (Figure 1), it was observed that *K. pneumoniae* (n = 24), *P. aeruginosa* (n = 20), *A. baumannii* (n = 11), and *E. coli* (n = 7) were the most prevalent bacteria in cultures isolated from ICU patients. Likewise, all bacteria except *Citrobacter freundii* had a carbapenemase. These results are similar to those obtained by Mayta-Barrios et al. (2021), who found a prevalence of 42.2% for *Enterobacteriaceae*, 32.9% for *P. aeruginosa*, and 24.9% for *Acinetobacter* spp. from cultures collected from different regions of Peru [25]. 

Table 3 shows the bacterial cultures identified by type of biological sample; it was observed that *P. aeruginosa* (n = 26), *E. coli* (n = 21), *K. pneumoniae* (n = 16), and *Burkholderia cepacia complex* (n = 16) had a higher frequency in aseptic urine samples. These results are similar to those reported by Spiess et al. (2022), who showed a higher prevalence of up to 39.2% for *K. pneumoniae* and 34.2% for *E. coli* [21]. Several studies mention that the high prevalence of these microorganisms in urinary tract infections is usually related to vulnerable patients with long stays in hospitals and previous use of broad-spectrum antibiotic therapy, in addition to risk factors such as age, prolonged hospitalization in the ICU, nephrological pathology, and other debilitating conditions such as immunosuppression identified in these patients [38]. It was observed that among the bronchial secretion samples, *K. pneumoniae* (n = 18) and *P. aeruginosa* (n = 12) were the most frequent. In addition, they presented resistance to both imipenem and meropenem. It is worth mentioning that Quintero and Varón (2022) found that 36.8% of *K. pneumoniae* isolates are multidrug-resistant, thus reaffirming that this infection is a major public health problem due to its association with mortality rates greater than 50%. In addition, in ICUs, there is a high risk of developing infections because they originate from the same microbiota of the patients, and long hospital stays and previous antibiotic treatment alter the microflora present, facilitating the excessive growth of pathogens [40].

Resistance to β-lactams has been extensively studied. In the case of *Enterobacteriaceae*, it results mainly from the expression of enzymes that inactivate antibiotics, beta-lactamases. In addition, new strains of resistant bacteria have been reported in *E. coli*, *Klebsiella oxytoca*, *Enterobacter*, *Serratia*, and *Salmonella* strains [1,7,41]. In the case of *Pseudomonas aeruginosa*, porin expulsion and loss pumps, which are resistance mechanisms different from β -lactamases, predominate; as for *Acinetobacter* spp., it has been mentioned that they have innate resistance mechanisms against multiple antimicrobials in their central genome [38,41]. The enzymes present in *Enterobacteriaceae* of the type KPC (*Klebsiella pneumoniae* carbapenemase), NDM (New Delhi Metallo-beta-lactamase), OXA (oxacillinases), IMP (imipenemases), and VIM (Verona integron-encoded metallo-beta-lactamase) are the most frequently detected worldwide [26,38], and due to the presence of COVID-19, in many South American countries there was an increase in the incidence of resistance to carbapenems, which can be attributed to the increase in the indiscriminate use of broad-spectrum antibiotics [8,42]. Spiess et al. (2022), when they analyzed the resistance mechanisms of Gram-negative bacteria, reported that 42.1% produced extended-spectrum beta-lactamase (ESBL), and these were more prevalent in *K. pneumoniae* strains; these results have sounded alarm bells at the regional level since in previous years, the prevalence did not exceed 30% in South America [21].

The carbapenemase present in *Enterobacteriaceae*, *P. aeruginosa*, and *A. baumannii* confers resistance on the antimicrobials imipenem or meropenem, as shown in Table 4. Various cultures of *P. aeruginosa* have been shown to have at least one of the three carbapenemases, which is of great concern and indicates that an epidemiological surveillance program should be continued. Likewise, as of 2020, the Ministry of Health of Peru has reported the appearance of a strain of *A. baumannii* of the NDM + OXA-58 type and two strains of *Klebsiella pneumoniae*, with double production of carbapenemases of the unusual type (KPC + NDM), and an unusual strain of *E. coli* with double production of carbapenemases (NDM + OXA-48) [8,11,26]; for this reason, it is important to carry out epidemiological surveillance to detect new unusual strains, as well as those with double or triple production of carbapenemases. 

Mayta-Barrios et al. (2021) observed that class B carbapenemases are the ones found at the greatest frequency, 62.7%, managing to identify the *bla*_NDM_ gene in 53 strains (44 strains of *Klebsiella pneumoniae* and 9 strains of *Escherichia coli*); secondly, they mentioned class D carbapenemases were found at a frequency of 24.9%, managing to identify the *bla*_OXA-48-like_ gene in 27 strains of *Acinetobacter* spp., and in third place was class A carbapenemases, for which the *bla*_KPC_ gene was identified with a frequency of 12.4%, [25]. The study also presented a global overview of the distribution of carbapenemases at the national level, while in the present research work, we identified the frequency of carbapenemases in a typical hospital in Trujillo, as shown in Table 5. It is worth mentioning that of the 66 cultures positive for carbapenemases, 28 were positive for the enzyme KPC (42.4%), 25 for NDM (37.9%), and 13 for OXA-48 (19.7%), with the class A and B carbapenemases being those that were most frequently found in isolated cultures. The results regarding the frequency of the type of carbapenemases found in the different bacterial groups (e.g., class A carbapenemases) differ from those of a similar study carried out in 2019 by the National Institute of Health (INS) of Peru, where it was reported that 185 nationally collected strains had class A, B, and D carbapenemase enzymes, with class B being the most frequent; they also mentioned genes such as KPC, NDM, IMP, VIM, OXA-24, and OXA-23 [24,25]. Similarly, in a study carried out in Spain during 2017, it was reported that among the class B carbapenemases, VIM-2 was the one that had the highest frequency, while among the class A carbapenemases, GES was the most frequently identified, and for class D, carbapenemases of the type OXA were the most prevalent [43].

The knowledge acquired in microbiology and our understanding of the patterns of resistance to antimicrobials is essential today since the emergence of new pathogens resistant to antimicrobials and especially to carbapenems has been observed. Therefore, it is necessary to provide adequate and timely information that will greatly support medical professionals in selecting appropriate treatments for patients, reduce health costs, and prevent the appearance of bacteria resistant to antimicrobials [21].

## 5. Conclusions

This descriptive study showed a high frequency of Gram-negative bacteria (*K. pneumoniae* and *P. aeruginosa*) with resistance to the carbapenems imipenem and meropenem in samples of bronchial secretions and aseptic urine from patients hospitalized in the Victor Lazarte Echegaray Hospital ICU during the pandemic. Of the 129 bacterial cultures studied, *K. pneumoniae* (31.6%), *P. aeruginosa* (26.3%), and *A. baumannii* (14.5%) were the most frequently isolated. Additionally, 24% of the *K. pneumoniae* cultures presented KPC, and 16% of the *P. aeruginosa* cultures presented at least one carbapenemase: NDM (n = 11), OXA-48 (n = 4), and KPC (n = 1). If the less frequently identified groups of bacteria with resistance to carbapenems are not controlled, they could cause serious mortality in the intensive care unit (ICU). The study findings show that greater control of antibiotic-resistant bacteria is needed. Finally, it is important to implement effective disinfection methods in ICUs. In addition, implementing an antibiotic policy that prevents the development of new antibiotic-resistant strains that compromise patients’ and health personnel’s health is also important. Future research is recommended to ensure updated information about the frequency of these bacteria in different hospitals in the city to minimize health risks.

## Figures and Tables

**Figure 1 medicina-59-01763-f001:**
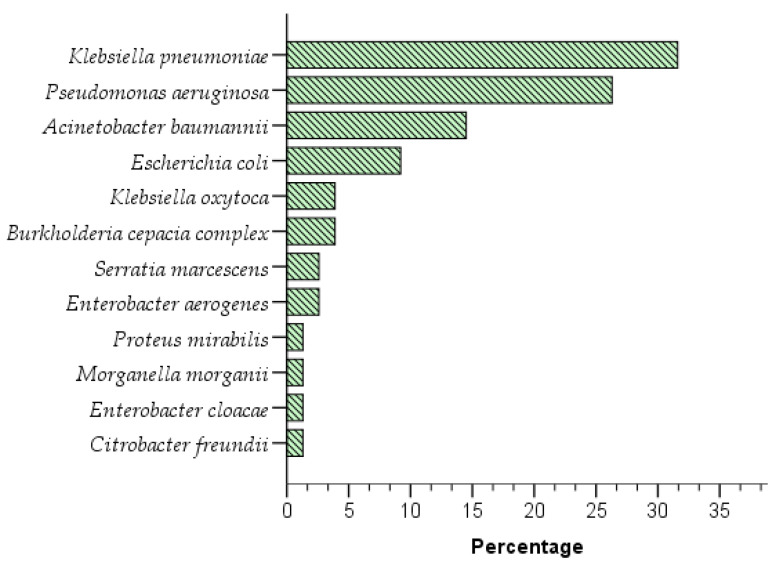
Distribution of bacteria isolated and identified by the automated AutoScan-4 system from biological samples taken from patients in the Víctor Lazarte Echegaray Hospital ICU from May 2021 to March 2022.

**Table 1 medicina-59-01763-t001:** Sociodemographic characteristics of ICU patients of the Victor Lazarte Echegaray Hospital.

	n = 129 (100.0%)	
	Frequency	Percentage
**Age**		
<40 years	10	7.8
≥40 years	119	92.2
**Gender**		
Female	61	47.3
Male	68	52.7

**Table 2 medicina-59-01763-t002:** Distribution of frequency of isolated bacterial cultures and resistance to carbapenemases according to the type of sample obtained from patients in the Víctor Lazarte Echegaray Hospital ICU from May 2021 to March 2022.

Type of Sample	Isolated Culture	Carbapenemase-Resistant Culture
	n (%)	n (%)
Bronchial secretion	60 (46.5)	43 (56.6)
Aseptic urine	41 (31.8)	19 (25.0)
Blood culture	13 (10.1)	2 (2.6)
Central venous catheter	5 (3.9)	2 (2.6)
Cerebrospinal fluid	3 (2.3)	3 (3.9)
Secretion of wound	2 (1.6)	2 (2.6)
Nasal discharge	1 (0.8)	1 (1.3)
Tracheal secretion	1 (0.8)	1 (1.3)
Pleural fluid	1 (0.8)	1 (1.3)
Sputum	1 (0.8)	1 (1.3)
Surgical wound	1 (0.8)	1 (1.3)
Total	129 (100)	76 (100)

**Table 3 medicina-59-01763-t003:** Distribution of frequency of bacteria isolated from aseptic urine samples and bronchial secretions in Víctor Lazarte Echegaray Hospital ICU patients from May 2021 to March 2022.

Bacteria	Bronchial Secretion	Aseptic Urine
	n (%)	n (%)
*Klebsiella pneumoniae*	18 (42.9)	3 (15.0)
*Pseudomonas aeruginosa*	12 (28.6)	5 (25.0)
*Acinetobacter baumannii*	5 (11.9)	2 (10.0)
*Escherichia coli*	2 (4.8)	4 (20.0)
*Enterobacter aerogenes*	1 (2.4)	0 (0.0)
*Klebsiella oxytoca*	2 (4.8)	1 (5.0)
*Serratia marcescens*	1 (2.4)	0 (0.0)
*Enterobacter cloacae*	1 (2.4)	0 (0.0)
*Burkholderia cepacia complex*	0 (0.0)	3 (15.0)
*Morganella morganii*	0 (0.0)	1 (5.0)
*Proteus mirabilis*	0 (0.0)	1 (5.0)
Total	42 (100.0)	20 (100.0)

**Table 4 medicina-59-01763-t004:** Distribution of bacteria resistant to meropenem and imipenem isolated from patients from the Víctor Lazarte Echegaray Hospital ICU.

Bacterial Culture	N (%)	Imipenem		Meropenem	
		R	S	R	S
*Klebsiella pneumoniae*	24 (31.6)	16	8	17	7
*Pseudomonas aeruginosa*	20 (26.3)	16	4	20	0
*Acinetobacter baumannii*	11 (14.5)	11	0	11	0
*Escherichia coli*	7 (9.2)	6	1	6	1
*Klebsiella oxytoca*	3 (3.9)	3	0	3	0
*Burkholderia cepacia complex*	3 (3.9)	2	1	3	0
*Enterobacter aerogenes*	2 (2.6)	2	0	2	0
*Serratia marcescens*	2 (2.6)	2	0	2	0
*Enterobacter cloacae*	1 (1.3)	1	0	1	0
*Citrobacter freundii*	1 (1.3)	1	0	1	0
*Morganella morganii*	1 (1.3)	1	0	1	0
*Proteus mirabilis*	1 (1.3)	0	1	1	0
Total	76 (100.00)	61	15	68	8

R = antibiotic-resistant; S = sensitive to antibiotics.

**Table 5 medicina-59-01763-t005:** Distribution of carbapenemase-producing bacteria isolated from patients from the Víctor Lazarte Echegaray Hospital ICU.

Bacteria	N (%)	KPC	NDM	OXA-48
*Klebsiella pneumoniae*	24 (36.4)	24	0	0
*Pseudomonas aeruginosa*	16 (24.2)	1	11	4
*Acinetobacter baumannii*	7 (10.6)	0	6	1
*Escherichia coli*	7 (10.6)	0	5	2
*Klebsiella oxytoca*	3 (4.5)	3	0	0
*Burkholderia cepacia complex*	3 (4.5)	0	1	2
*Enterobacter aerogenes*	2 (3.0)	0	0	2
*Serratia marcescens*	1 (1.5)	0	0	1
*Enterobacter cloacae*	1 (1.5)	0	1	0
*Morganella morganii*	1 (1.5)	0	1	0
*Proteus mirabilis*	1 (1.5)	0	0	1
Total	66 (100.00)	28	25	13

## Data Availability

No applicable.

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
