# Peer review of "Carbapenemase-Producing Bacteria Isolated from ICU Patients of a Peruvian Government Hospital during the COVID-19 Pandemic: A Descriptive Analysis"

_medicina, 2023, doi:10.3390/medicina59101763_

Round 1

Reviewer 1 Report

The manuscript submitted by García-Cedrón et al. is dedicated to the study of carbapenems-resistant (Imipenem- and Meropenem-resistant) bacteria isolated from a Peruvian hospital during COVID-19. The authors presented their frequency and identified them using the automated system AutoScan-4, as well as the RESISIT-3 O.K.N K-SET cassettes used for carbapenemase.

I do believe that the topic and the results presented are interesting. I highly recommend that this article be published in the journal in its actual version.

The authors could improve the manuscript English!

Author Response

We appreciate your comments. It will be taken into account with respect to improving English.

Reviewer 2 Report

In the manuscript Titled “Carbapenemase-producing bacteria isolated from ICU patients of a Peruvian Government Hospital during COVID-19: A descriptive analysis”, the authors studied the frequency and type of carbapenemases present in Gram-negative bacteria isolated from patients hospitalized in the ICU area of the Victor Lazarte Eche garay Hospital in Peru.

They isolated the bacteria from a wide range of samples and studied their resistance pattern against carbapenem antibiotics. They further study the occurrence of three different carbapenemases viz. KPC, OXA, and NDM. Though there is nothing new in the work, the data has a piece of significant information to add to the field of antimicrobial resistance. Furthermore, the manuscript is designed well and the language is satisfactory.

There are a few comments which in my opinion should be addressed before going any further-

1)     Line 138-139: ‘Due to the prolonged use of broad-spectrum antibiotics generating multi-drug-resistant bacteria’ Consider removing this statement. there could be many other reasons as well for the emergence and dissemination of MDR bacteria.

2)     Line 142: ‘129 bacterial cultures isolated from samples from UCI’ What is UCI?

3)     Line 182: ‘To identify extended-spectrum β-lactams’ It should be extended-spectrum β-lactamases.

4)     Line 200-201- Which version of CLSI was referred to?

5)     Table-2: The high number of bacteria were isolated from bronchial secretion and urine. But It could be seen in the table that isolates obtained starting from cerebrospinal fluid to surgical wound have 100 percent resistance towards carbapenems. what is the reason behind it? What is the significance of this pattern in your study?

6)     Line 260-261: What is MDM? Is it NDM or something else?

The manuscript required a thorough check to correct the spelling and grammatical errors.

Author Response

We appreciate your comments. The responses to their observations are attached.

Reviewer 3 Report

This manuscript offers a window on the prevalence and antibiotics resistance of several types of Carbapenemase-producing bacterial strains isolated from ICU patients of a Peruvian Government Hospital during COVID-19. Although this study is interesting, I do not recommend accepting this paper in its current state, for the following reasons:

·      A major flaw in this study is relying only on an automated system for species identification of the bacterial isolates. The authors could have used PCR to confirm the identity of the isolates included in the study. 

·      The authors should have identified the carbapenemase genes in all isolates by molecular means, and corelated the phenotypic results with the types of genes identified.

·      The author should explain why they did choose to only investigate the presence of KPC, NDM, and OXA-48 

·      The authors could have provided more data about the resistance of the isolates to other antimicrobials

Minor issues

The authors should use the proper nomenclature for gene and bacteria names. Throughout the manuscript, the blaOXA-48 and blaKPC genes should be written as italic with the OXA-48 and KPC part as subscript as follows: blaKPC and blaOXA-48

There is no need to continue writing the full genus name of Acinetobacter baumannii throughout the document. Instead use the abbreviated genus with the full species name; A. baumannii

Minor typos

Author Response

(The authors gave the same response as above.)

Round 2

Reviewer 3 Report

First, I would like to thank the authors for their response. Unfortunately, I don't think there is enough data to warrant a full article publication. Also, the method used for bacterial species identification is a major limitation of this manuscript. 

Just minor issues